# In Silico Binding of 2-Aminocyclobutanones to SARS-CoV-2 Nsp13 Helicase and Demonstration of Antiviral Activity

**DOI:** 10.3390/ijms24065120

**Published:** 2023-03-07

**Authors:** Thahani S. Habeeb Mohammad, Yash Gupta, Cory T. Reidl, Vlad Nicolaescu, Haley Gula, Ravi Durvasula, Prakasha Kempaiah, Daniel P. Becker

**Affiliations:** 1Department of Chemistry and Biochemistry, Loyola University Chicago, Chicago, IL 60660, USA; 2Department of Medicine, Division of Infectious Diseases, Mayo Clinic, Jacksonville, FL 32224, USA; 3Howard T. Ricketts Laboratory, Department of Microbiology, The University of Chicago, Chicago, IL 60637, USA

**Keywords:** peptidomimetics, HTS, high-throughput in silico screen, coronavirus variants, nonstructural protein 13, Nsp13 helicase, antiviral, cyclobutanone, enzyme inhibitor, transition state mimetic, carbonyl hydrate, computer-aided drug design, SAR

## Abstract

The landscape of viral strains and lineages of SARS-CoV-2 keeps changing and is currently dominated by Delta and Omicron variants. Members of the latest Omicron variants, including BA.1, are showing a high level of immune evasion, and Omicron has become a prominent variant circulating globally. In our search for versatile medicinal chemistry scaffolds, we prepared a library of substituted ɑ-aminocyclobutanones from an ɑ-aminocyclobutanone synthon (**11**). We performed an in silico screen of this actual chemical library as well as other virtual 2-aminocyclobutanone analogs against seven SARS-CoV-2 nonstructural proteins to identify potential drug leads against SARS-CoV-2, and more broadly against coronavirus antiviral targets. Several of these analogs were initially identified as in silico hits against SARS-CoV-2 nonstructural protein 13 (Nsp13) helicase through molecular docking and dynamics simulations. Antiviral activity of the original hits as well as ɑ-aminocyclobutanone analogs that were predicted to bind more tightly to SARS-CoV-2 Nsp13 helicase are reported. We now report cyclobutanone derivatives that exhibit anti-SARS-CoV-2 activity. Furthermore, the Nsp13 helicase enzyme has been the target of relatively few target-based drug discovery efforts, in part due to a very late release of a high-resolution structure accompanied by a limited understanding of its protein biochemistry. In general, antiviral agents initially efficacious against wild-type SARS-CoV-2 strains have lower activities against variants due to heavy viral loads and greater turnover rates, but the inhibitors we are reporting have higher activities against the later variants than the wild-type (10–20X). We speculate this could be due to Nsp13 helicase being a critical bottleneck in faster replication rates of the new variants, so targeting this enzyme affects these variants to an even greater extent. This work calls attention to cyclobutanones as a useful medicinal chemistry scaffold, and the need for additional focus on the discovery of Nsp13 helicase inhibitors to combat the aggressive and immune-evading variants of concern (VOCs).

## 1. Introduction

In December 2019, a new species of coronavirus was identified as severe acute respiratory syndrome coronavirus 2 (SARS-CoV-2) [1], the causative agent of the global pandemic of coronavirus disease in 2019 (COVID-19) [2]. This highly contagious and often fatal coronavirus has caused over 755 million COVID-19 cases globally, resulting in over 6.8 million deaths, while in the United States, over 101 million COVID-19 cases have been reported, resulting in over 1.1 million deaths (accessed 13 February 2023) [3]. The reciprocal evolution and continued emergence of new SARS-CoV-2 variants raise concerns regarding the loss of efficacy of the few antivirals we have at our disposal that can keep this pandemic under control. The emerging SARS-CoV-2 Delta and Omicron variants with multiple mutations have been associated with increased transmissibility and rapid turnover rates [4]. Targeting the same proteins, the main protease (Mpro) in particular, increases the chance of cross-resistance through resistance-conferring polymorphisms that thwart drug discovery and development efforts [5]. Diversification of the targets is the key to developing an effective multi-drug therapy [6]. The small genomes of viruses do not offer many therapeutic options because they are heavily dependent on the host system, and many protein components have roles that are poorly understood [7]. Even with so much focus from various fields of science during the SARS-CoV-2 pandemic, mysteries remain to be unraveled regarding SARS-CoV-2 [8]. There have been some rushed efforts to repurpose or develop therapeutics and perform trials with unprecedented drug/patient loads globally [9]. A continued effort is warranted to explore new key targets for effective drug development against SARS-CoV-2 and new variants [10]. Target-based drug development is the most rapid pipeline to deliver potent molecules even from weaker initial hits [11].

We have a longstanding interest in cyclobutanones [12,13] as peptidomimetics and potentially privileged medicinal chemistry scaffolds. Cyclobutanones provide a degree of conformational rigidity, while the strain inherent in the 4-membered ring makes the ketone carbonyl more electrophilic than an unstrained ketone. This electrophilic carbonyl can reversibly inhibit serine proteases, while the hydrated form of the electrophilic carbonyl provides a transition-state mimetic for inhibiting metalloenzymes. Given that diseases involving inappropriate proteolytic activity may be treated by selectively inhibiting an implicated protease enzyme [14], we envisioned cyclobutanones as a privileged structure for protease inhibitors offering a compact, substitutable scaffold with the embedded electrophilic carbonyl. Early on, peptidic cyclobutanones were shown to inhibit the serine protease elastase [15], while cyclobutanones targeting isopenicillin N-synthase have also been reported [16]. Cyclobutanone inhibitors of both serine- and metallo-β-lactamases have also been prepared [17], and more recently an X-ray crystal structure of a class B metallo-β-lactamase SPM-1 with a hydrated cyclobutanone coordinated to the di-zinc metal center was solved [18]. We revealed the X-ray crystal structure of a cyclobutanone-based inhibitor covalently bound to a key esterase from the Gram-negative pathogenic coccobacillus *Francisella tularensis* [19].

In the present study, we performed a virtual screen of a library of synthesized 2-aminocyclobutanone derivatives as well as related virtual derivatives against seven non-structural protein targets of therapeutic interest in SARS-CoV-2, including 3-chymotrypsin-like protease (3CLPro), papain-like protease (PLpro), RNA-directed RNA polymerase (RdRp), nonstructural protein 13 (Nsp13) helicase, nonstructural uridylate-specific endoribonuclease (NendoU), exoribonuclease (ExoN), and 2′-O-methyltransferase (2′-O-MT). We identified 2-aminocyclobutanone hits for PLpro, 3CLPro, RdRp, and Nsp13 helicase, but the most robust and interesting hit was *para*-toluenesulfonyl (tosyl)-D-Phe-aminocyclobutanone hydrate versus Nsp13 helicase. Nsp13 helicase, designated as nonstructural protein 13 (Nsp13) of SARS-CoV-2, is an enzyme found in the viral replication transcription complex in coronaviruses, and it has been shown to have nucleoside triphosphate hydrolase NTPase activity and RNA helicase activity [20]. Helicases are motor proteins that utilize the free energy from NTP hydrolysis in unwinding helical duplex DNA and RNA as well as folded viral genomic RNA with intramolecular RNA-RNA interactions [21]. There are reports that host helicases also have indispensable roles in viral replication [22,23]. Helicase activity is vital in viral genome replication, recombination, transcription, and repair [22,23]. Significantly, coronavirus helicase is the most conserved protein across CoV genera, and the SARS-CoV-2 Nsp13 helicase has a 99.8% sequence identity to SARS-CoV helicase [24]. This high level of conservation shows the importance of the Nsp13 helicase for the survival of coronaviruses [25]. X-ray crystal structures of MERS-CoV and SARS-CoV Nsp13 helicases revealed that coronavirus helicases are highly similar in structure [26]. Helicases consist of multiple domains including an *N*-terminal Zn (II) binding domain (ZBD), a hinge domain (stalk), and a 1B domain; together, they connect the helicase core (SF1) [25]. Ting Shu and co-workers demonstrated the magnesium dependency of SARS-CoV-2 Nsp13 helicase and they reported the inhibition by bismuth of both NTPase and RNA helicase activities of SARS-CoV-2 Nsp13 [20]. Due to high phylogenetic conservation, a known SARS-CoV helicase inhibitor cepharanthine was demonstrated to have modest (IC_50_ = 0.4 mM) anti-SARS-CoV-2 activity [24]. Figure 1 illustrates the structure of SARS-CoV-2 Nsp13 helicase, including the adenosine triphosphate (ATP)-binding region, where the 2-aminocyclobutanone derivatives described herein are predicted to bind.

To provide a readily available modular α-amino cyclobutanone surrogate, we reported the synthesis of an α-aminocyclobutanone synthon that can provide a range of substituted derivatives, including α-benzamide, α-thiourea, α-urea, and α-sulfonamide cyclobutanones, in a single pot [28]. Following this method, we synthesized a diverse set of α-aminocyclobutanones prioritized by computer-aided drug design (CADD) screenings against SARS-CoV-2 Nsp13 helicase. As noted above, we performed a virtual screen of a library of synthesized 2-aminocyclobutanone derivatives as well as virtual related derivatives against seven non-structural protein targets of therapeutic interest in SARS-CoV-2, including 3CLPro, PLpro, RdRp, Nsp13 helicase, NendoU, ExoN, and 2′-O-MT. We identified 2-aminocyclobutanone hits for PLpro, 3CLPro (Mpro), RdRp, and Nsp13 helicase, but the most robust and interesting hit was that of tosyl-D-Phe-aminocyclobutanone hydrate versus SARS-CoV-2 Nsp13 helicase. We were expecting 3CLPro hits from this library biased toward protease inhibitors. We opted to pursue the Nsp13 helicase hit given the greater novelty and druggability of the chemical structure, and to focus on the less explored helicase target versus the more familiar 3CLPro (Mpro) target. Furthermore, the tosyl-D-Phe-aminocyclobutanone analog was physically in our synthesized compound library, and docking hypothetical related analogs provided improved docking scores and a forward direction for additional analogs. We confirmed in silico inhibitory potencies with predicted strong target-ligand interactions against SARS-CoV-2 Nsp13 helicase, and subsequently demonstrated anti-viral efficacies of these α-amino cyclobutanones that we report herein, revealing a new class of small molecule SARS-CoV-2 Nsp13 helicase inhibitors with antiviral activity, thus confirming that cyclobutanones offer a useful scaffold for the construction of enzyme inhibitors.

The High-Throughput Virtual (HTV) screening was performed in the Glide module of the Schrödinger suite, a grid-based ligand docking utilizing free energy calculations [9]. We developed structure-based pharmacophore models for each target hit pool using PharmaGist [29] and the Molecular Operating Environment (MOE) [30]. Hits identified through our CADD screenings, which involved screening both cyclobutanone carbonyl and hydrated forms as well as both epimers at the cyclobutanone alpha-amino-bearing carbon, provided comprehensive ligand binding interaction profiles. Pharmacophore models combined with the ligand binding profiles enabled us to generate a series of hit-inspired analogs through functional group modifications. Prospective cyclobutanone analogs were virtually screened against SARS-CoV-2 Nsp13 helicase, guiding further synthesis. Top-scoring cyclobutanone derivatives were synthesized, and in vitro activities of these hits were validated through biochemical and cellular assays.

## 2. Results

### 2.1. Synthesis

Ap-Toluenesulfonyl chloride (TsCl) was reacted with amino acids in a base-mediated coupling reaction using Na_2_CO_3_ following a published general procedure [31]. Sulfonamide intermediates of free carboxylic acids (**12a**–**c**) were then reacted in a propylphosphonic anhydride (T3P) coupling reaction with the 2-aminocyclobutanone synthon **11** with T3P and N-methylmorpholine (NMM) in ethyl acetate (EA) to give acetal intermediates, which were hydrolyzed under general reaction conditions (1 M HCl, acetone, H_2_O, rt to 40 °C) and yielded the final p-toluenesulfonamide-amino acid-cyclobutanone analogs **1**, **4**, and **6** (Figure 1).

For the Cbz-amino acid cyclobutanone derivatives, the Cbz protected amino acids were coupled with acetal **11** using T3P as the coupling agent, and selective hydrolysis of the acetal while preserving the Cbz. This selective hydrolysis was accomplished with a 40% *v*/*v* acetic acid solution in tetrahydrofuran (THF, 40% *v*/*v*) and water at elevated temperatures up to 60 °C providing Cbz cyclobutanone analogs **2**, **3**, **7**, and **9** (Figure 2).

4-Methoxybenezene D-valine sulfonamide cyclobutanone **8** was synthesized via *N*-sulfonylation, ester hydrolysis, and coupling reactions utilizing D-valine *tert*-butyl ester HCl salt with 4-methoxybenzenesulfonyl chloride in pyridine yielding the corresponding sulfonamide. Deprotection of the *tert*-butyl ester was achieved by stirring the ester with trifluoroacetic acid (TFA) in dichloromethane (DCM). The cyclobutanone analog **8** was then prepared by reacting 2-aminocyclobutanone synthon **11** with ((4-methoxyphenyl) sulfonyl)-D-valine (**14**) under T3P coupling conditions followed by hydrolysis as shown in Figure 3.

The cyclobutanone isostere-containing analogs **5** and **10** were prepared by coupling tosyl-L-Phe with synthon **11** using T3P and NMM in EA followed by hydrolysis (Figure 4).

For the synthesis of cyclobutanone derivatives **1**, **4,** and **6,** as illustrated in Figure 1 above, the requisite tosyl amino acids **12a–c** were prepared following the general method reported by Misra and co-workers [32]. Sodium carbonate (179.8 mg, 1.45 mmol) was added to a vigorously stirring solution of the amino acid (1.21 mmol) in DI water (1.5 mL). Once all the solutes were dissolved, the solution was cooled to 0 °C and the respective aryl sulfonyl chloride (1.45 mmol) was added to the stirring reaction mixture in four portions over 1 h. Then the reaction was allowed to warm up to rt and the slurry was stirred at rt for 4 h or until completion. The reaction progress was monitored by TLC (methanol/methylene chloride = 10/90). Once completed, the mixture was acidified using 2 N HCl until the pH of the solution was reduced to 2. Crystals formed during the acidification were filtered and washed with a pH 2.2 buffer. Final pure products (**12a**–**c**) were dried under a high vacuum. Both the proton and carbon NMR spectra of pure products **12a**–**c** matched the reported characterization data as indicated.

To a solution of substituted *N*-benzene sulfonamide, amino acids **12a**–**c** (1 eq) and acetal **11** (1.2 eq) in ethyl acetate (0.2 M), *N*-methyl morpholine (4.0 eq) was added. Then T3P (2.5 eq) was added to the above mixture as a solution in ethyl acetate (purchased as ≥ 50 weight % in EA). The reaction mixture was stirred under N_2_ at rt to 40 °C for 18 h or until complete consumption of protected carboxylic acid intermediates was determined by an HPLC analysis. The reaction was quenched by adding water (3 mL) and then the organic product was extracted using ethyl acetate (3 × 3 mL). The combined organic layers were washed successively with water (3 × 3 mL) and 1N HCl (3 mL) then dried over Na_2_SO_4_. The solvent was removed by evaporation under reduced pressure, providing the corresponding acetal intermediate. Without further purification, crude acetal (1 eq, final concentration was 0.1 M) was subjected to hydrolysis conditions, wherein the crude mixture was dissolved in acetone, water (10% *v*/*v*), and 1N HCl (30% *v*/*v*). Then the reaction mixture was stirred at 40 °C overnight with periodic HPLC monitoring. Upon completion, the organic product was extracted using ethyl acetate (3 × 3 mL) or methylene chloride for highly water-soluble analogs, and the combined organic layers were dried over Na_2_SO_4_. The solvent was evaporated under reduced pressure, and the crude mixture was purified by column chromatography to afford the corresponding amino acid-derived cyclobutanone (**1**, **4**, **6**).

2S)-2-((4-Methylphenyl)-sulfonamido)-*N*-(2-oxocyclobutyl)-3-phenylpropanamide (**1**). The *N*-tosyl-carboxylic intermediate **12a** (50.0 mg, 0.129 mmol) was used in the reaction. The crude mixture of **1** was purified by column chromatography eluting with a gradient of ethyl acetate/hexane (20/80) and ethyl acetate/hexane (40/60) followed by recrystallization from hot EA to afford compound **1** (9.8 mg, 20%) as a white crystalline solid. ^1^H NMR (500 MHz, CDCl_3_) δ 7.57–7.44 (m, 2H), 7.25–7.15 (m, 5H), 6.96–6.87 (m, 2H), 6.72 (d, *J* = 8.3 Hz, 1H), 4.97 (dtt, *J* = 10.1, 8.2, 2.1 Hz, 1H), 4.85 (d, *J* = 6.6 Hz, 1H), 3.85 (dt, *J* = 7.3, 6.4 Hz, 1H), 3.01–2.89 (m, 3H), 2.88 (dddd, *J* = 17.4, 9.9, 4.7, 2.3 Hz, 1H), 2.43 (s, 3H), 2.46–2.35 (m, 1H), 1.86 (dtd, *J* = 11.1, 9.7, 8.3 Hz, 1H). HRMS (ESI): Calcd for (MH^+^) C_20_H_23_N_2_O_4_S: 387.1373, found 387.1366.

2S)-3-(1H-Indol-3-yl)-2-((4-methylphenyl)-sulfonamido)-N-(2-oxocyclobutyl) propanamide (**4**). The *N*-tosyl-carboxylic intermediate **12b** (50.0 mg, 0.139 mmol) was used in the reaction. The crude mixture of **4** was purified by column chromatography on a Teledyne Isco Rf Flash chromatography unit eluting with a gradient of ethyl acetate/hexane (20/80), ethyl acetate/hexane (30/70), and ethyl acetate/hexane (40/60) to afford compound **4** (12.4 mg, 21%) as an off-white solid: mp 200–201 °C.^1^H NMR (500 MHz, acetonitrile-*d*_3_) δ 9.07 (NH, s, 1H), 7.48–7.41 (m, 2H), 7.41–7.32 (m, 2H), 7.16–7.09 (m, 3H), 7.10 (NH, d, *J* = 7.5 Hz, 1H), 7.07–6.96 (m, 2H), 5.82 (NH, s, 1H), 4.68 (dtt, *J* = 10.2, 8.0, 2.2 Hz, 1H), 3.90 (dd, *J* = 8.1, 5.7 Hz, 1H), 3.14–3.06 (m, 1H), 2.95–2.73 (m, 3H), 2.35 (s, 3H), 2.21–2.10 (m, 1H), 1.82 (tt, *J* = 10.5, 8.6 Hz, 1H). ^13^C NMR (126 MHz, CD_3_CN) δ 205.61, 170.72, 143.47, 136.61, 136.41, 129.35, 127.10, 126.57, 126.51, 124.06, 121.41, 118.88, 111.33, 108.97, 63.62, 63.60, 57.04, 41.40, 41.34, 28.72, 28.58, 20.60, 20.58, 18.18, 18.11. HRMS (ESI): Calcd for (MH^+^) C_22_H_24_N_3_O_4_S: 426.1482, found 426.1472.

2R)-2-((4-Methylphenyl) sulfonamido)-*N*-(2-oxocyclobutyl)-3-phenylpropanamide (**6**). The *N*-tosyl-carboxylic intermediate **12c** (50.0 mg, 0.129 mmol) was used in the reaction. The crude mixture of **6** was purified by column chromatography eluting with a gradient of ethyl acetate/hexane (20/80) and ethyl acetate/hexane (30/70) to afford compound **6** (18.1 mg, 36.2%) as a white solid: mp 190–192 °C. ^1^H NMR (500 MHz, CDCl_3_) δ 7.57–7.48 (m, 2H), 7.24–7.14 (m, 5H), 6.95–6.86 (m, 2H), 6.75 (NH, d, *J* = 8.3 Hz, 1H), 4.98 (dtt, *J* = 10.2, 8.3, 1.9 Hz, 0.5H), 4.87 (NH, d, *J* = 6.6 Hz, 0.5H), 4.82 (NH, d, *J* = 6.8 Hz, 0.5H), 4.78–4.68 (m, 0.5H), 3.87 (dtd, *J* = 16.8, 7.0, 6.1 Hz, 1H), 3.07–2.93 (m, 1.5H), 2.96–2.91 (m, 1H), 2.93–2.85 (m, 1H), 2.83 (dd, *J* = 14.1, 5.8 Hz, 0.5H), 2.44 (s, 1.5H), 2.43 (s, 1.5H), 2.43–2.36 (m, 0.5H), 2.39–2.31 (m, 0.5H), 2.16 (dddd, *J* = 11.2, 10.3, 8.9, 8.0 Hz, 0.5H), 1.86 (dtd, *J* = 11.2, 9.7, 8.4 Hz, 0.5H). ^13^C NMR (126 MHz, CDCl_3_) δ 204.49, 204.40, 170.23, 169.93, 144.15, 144.09, 135.42, 135.30, 134.92, 134.71, 129.95, 129.90, 129.26, 129.14, 129.04, 127.42, 127.39, 127.21, 127.19, 77.27, 77.02, 76.77, 64.30, 63.68, 57.62, 57.29, 42.33, 42.13, 38.19, 37.74, 21.61, 21.59, 19.43, 18.90.

(2R)-2-((4-Methoxyphenyl) sulfonamido)-3-methyl-*N*-(2-oxocyclobutyl) butanamide (**8**). D-Valine *tert*-butyl ester HCl salt (100 mg, 0.477 mmol) was dissolved in anhydrous pyridine (1.0 mL) and stirred until all the solutes were dissolved. The reaction mixture was cooled to 0 °C in an ice bath, and 4-methoxybenzenesulfonyl chloride (98.5 mg, 0.477 mmol) was added to the stirring reaction mixture. Then the reaction was allowed to warm up to rt and stirred at rt for 24 h with periodic monitoring using HLPC. Upon completion, the reaction mixture was diluted with methylene chloride (3 mL), the organic layer was washed successively with water (3 mL), 1N HCl (10 × 3 mL) and brine (3 mL), and then dried over Na_2_SO_4_. The solvent was removed by evaporation under reduced pressure, leaving behind a pale, yellow solid. Without further purification, tosyl-D-valine *tert* butyl ester was subjected to hydrolysis conditions. To a solution of crude tosyl-D-valine *tert* butyl ester (64.7 mg, 0.188 mmol) in methylene chloride (0.94 mL), TFA (282 µL, 30% *v*/*v*) was added under nitrogen. The mixture was stirred at room temperature overnight and monitored by TLC (EA/hexane = 50/50). The solvent was evaporated under reduced pressure. To the resultant residue, toluene (2 mL) was added to enable the formation of an azeotrope. The excess TFA was removed by evaporation on a rotary evaporator at 50 °C, leaving behind a yellow crystalline solid (52.6 mg, 97%). HPLC purity of product **8** was 97%. Proton and carbon NMR spectra of the ((4-methoxyphenyl) sulfonyl)-D-valine (**14**) matched the published data [29]. Compound **8** was synthesized from intermediate **14** (30 mg, 0.104 mmol) following the general procedure of the T3P coupling reactions of *N*-Ts amino acids with 2-aminocyclobutanone synthon **11,** and the acetal intermediate was subjected to hydrolysis using general reaction conditions (1 M HCl, acetone, H_2_O) and stirred at 40 °C for 18 h. The crude mixture of **8** was purified by column chromatography using ethyl acetate/hexane (60/40) to afford the p-methoxyphenylsulfonyl-D-valine cyclobutanone analog **8** (15.5 mg, 42%) as a white solid: mp 179–181. ^1^H NMR (500 MHz, CDCl_3_) δ 7.84–7.77 (m, 2H), 7.04–6.96 (m, 2H), 6.51 (dd, *J* = 16.8, 7.7 Hz, 1H), 5.08 (dd, *J* = 13.7, 7.8 Hz, 1H), 4.89–4.76 (m, 1H), 3.90 (s,1.5H), 3.89 (s, 1.5H), 3.48 (ddd, *J* = 16.5, 7.8, 5.1 Hz, 1H), 2.99–2.87 (m, 2H), 2.43–2.30 (m, 1H), 2.17–2.06 (m, 1H), 2.01–1.94 (m, 0.5H), 1.87 (dtd, *J* = 11.0, 9.6, 8.1 Hz, 0.5H), 0.87 (dd, *J* = 6.9, 3.2 Hz, 3H), 0.81 (t, *J* = 6.7 Hz, 3H). ^13^C NMR (126 MHz, CDCl_3_) δ 204.22, 170.37, 170.30, 163.28, 130.53, 129.59, 114.37, 64.08, 63.90, 61.82, 61.80, 55.68, 42.29, 42.21, 31.11, 31.05, 19.19, 19.04, 17.15, 17.12. HRMS (ESI): Calcd for (MH^+^) C_16_H_23_N_2_O_5_S: 355.1322, found 355.1309.

S(S)-*N*-(2-hydroxy-2-methylpropyl)-2-((4-methylphenyl) sulfonamido)-3-phenylpropanamide (**10**). Tosyl-L-phenylalanine (50 mg, 0.129 mmol) was reacted with dimethylethanolamine (14.4 µL, 0.155 mmol) following the general T3P coupling procedure and stirred at 50 to 60 °C for 6 days until completion. The crude product (oil) of **10** was crystallized from hot EA/hexane = 20/80. When the EA/hexane solution was cooled to rt the product became an oil. Therefore, it is important to maintain the temperature of the solution to stay between 50 to 60 °C to allow a slow crystallization of the final product. Recrystallization provided compound **10** (29 mg, 58%) as clear needle-like crystals: mp 131–133 °C. ^1^H NMR (500 MHz, CDCl_3_) δ 7.53–7.47 (m, 2H), 7.23–7.14 (m, 5H), 6.96–6.90 (m, 2H), 6.74 (s, 1H), 4.93 (d, *J* = 6.2 Hz, 1H), 3.84 (dt, *J* = 7.8, 6.1 Hz, 1H), 3.29 (dd, *J* = 13.7, 6.9 Hz, 1H), 3.12 (dd, *J* = 13.7, 5.5 Hz, 1H), 2.97 (dd, *J* = 14.0, 5.9 Hz, 1H), 2.91 (dd, *J* = 14.0, 7.7 Hz, 1H), 2.43 (s, 3H), 1.16 (d, *J* = 2.2 Hz, 6H). ^13^C NMR (126 MHz, CDCl_3_) δ 170.92, 144.05, 135.19, 135.16, 129.87, 129.09, 129.04, 127.37, 127.23, 70.85, 58.06, 50.30, 38.29, 27.17, 27.10, 21.58. HRMS (ESI): Calcd for (MH^+^) C_20_H_27_N_2_O_4_S: 391.1686, found 391.1680.

S(S)-*N*-((1-hydroxycyclopropyl) methyl)-2-((4-methylphenyl) sulfonamido)-3-phenylpropanamide (**5**). Tosyl-L-phenylalanine **12a** (50 mg, 0.129 mmol) was reacted with 1-(aminomethyl) cyclopropanol (14.4 µL, 0.155 mmol) following the general T3P coupling procedure and stirred at 50 °C for 4 days. The crude mixture of **5** was purified by column chromatography on a Teledyne Isco Rf Flash chromatography unit eluting with a gradient of ethyl acetate/hexane (20/80) and ethyl acetate/hexane (30/70). Appropriate fractions were combined and concentrated to give a white solid which was then subjected to recrystallization. The product was dissolved in a minimum amount of hot methylene chloride and hexane was added dropwise until cloudiness was observed. While cooling needle-like crystals formed providing compound **5** (13.6 mg, 27%). mp 131–133 °C. ^1^H NMR (500 MHz, CDCl_3_) δ 7.51 (dd, *J* = 8.1, 2.4 Hz, 2H), 7.25 (d, *J* = 13.9 Hz, 1H), 7.21 (dd, *J* = 12.2, 7.2 Hz, 5H), 6.94 (d, *J* = 7.1 Hz, 2H), 6.75 (NH, d, *J* = 6.3 Hz, 1H), 4.96 (NH, d, *J* = 5.7 Hz, 1H), 3.86–3.80 (m, 1H), 3.39 (ddd, *J* = 14.3, 6.8, 2.4 Hz, 1H), 3.30–3.22 (m, 1H), 3.20 (OH, s, 1H), 2.95 (ttd, *J* = 14.1, 10.3, 8.5, 4.4 Hz, 2H), 2.43 (d, *J* = 2.4 Hz, 3H), 0.79 (s, 2H), 0.64–0.56 (m, 1H), 0.56–0.50 (m, 1H). ^13^C NMR (126 MHz, CDCl_3_) δ 171.13, 144.15, 135.17, 135.03, 129.88, 129.09, 129.06, 127.41, 127.26, 58.15, 55.43, 47.73, 38.32, 21.60, 12.84, 12.48.

For the synthesis of Cbz protected amino acid cyclobutanone analogs **2**, **3**, **7**, and **9**. As illustrated above in Figure 2, Cbz-protected amino acids (Phe, Try, Val, 50 mg) were coupled with 2-aminocyclobutanone synthon 11 following the general T3P coupling reaction procedure. The resulting acetal intermediates **13a**–**d** were subjected to hydrolysis without further purification. To a solution of crude acetal product (1 eq, concentration = 0.1 M) in THF (40% *v*/*v*), water (20% *v*/*v*) and acetic acid were added, and the reaction mixture was stirred at 60 °C overnight or until deemed complete by HPLC. Upon completion, excess acetic acid was neutralized by adding a solution of saturated Na_2_CO_3_ until a pH of 8 was achieved and the organic product was extracted with methylene chloride (3 × 3 mL). The combined organic layers were dried over Na_2_SO_4,_ and the solvent was evaporated under reduced pressure. The crude mixture was purified by column chromatography to afford the corresponding Cbz-protected amino acid-derived cyclobutanones (**2**, **3**, **7**, **9**).

Benzyl ((2R)-1-oxo-1-((2-oxocyclobutyl) amino)-3-phenylpropan-2-yl) carbamate (**2**). The crude mixture of **2** was purified by column chromatography eluting with a gradient of ethyl acetate/hexane (20/80) and ethyl acetate/hexane (30/70). Appropriate fractions were combined and concentrated to give a white solid (93% HPLC purity) which was then subjected to recrystallization. The product was dissolved in a minimum amount of methylene chloride and hexane was added dropwise until cloudiness was observed. Colorless crystals were formed providing compound **2** (16.1 mg, 26%): mp 98–100 °C. ^1^H NMR (500 MHz, CDCl_3_) δ 7.41–7.24 (m, 8H), 7.21 (t, *J* = 6.0 Hz, 2H), 6.35 (d, *J* = 7.6 Hz, 0.5H), 6.20 (d, *J* = 8.0 Hz, 0.5H), 5.32 (s, 0.5H), 5.27 (s, 0.5H), 5.10 (s, 2H), 4.92 (q, *J* = 9.0 Hz, 0.5H), 4.81 (q, *J* = 8.7 Hz, 0.5H), 4.42 (q, *J* = 7.3 Hz, 1H), 3.17–3.11 (m, 1H), 3.06 (td, *J* = 13.5, 12.9, 7.6 Hz, 1H), 2.99–2.85 (m, 2H), 2.39 (dq, *J* = 15.8, 9.7 Hz, 1H), 2.06–2.03 (m, 0.5H), 1.88–1.81 (m, 0.5H). ^13^C NMR (126 MHz, CDCl_3_) δ 204.33, 170.65, 170.49, 155.97, 136.02, 136.02, 129.39, 129.37, 128.84, 128.59, 128.30, 128.29, 128.09, 127.22, 67.23, 64.05, 63.76, 56.03, 42.21, 42.12, 38.56, 38.19, 19.42, 19.22.

Benzyl ((2S)-3-methyl-1-oxo-1-((2-oxocyclobutyl) amino) butan-2-yl) carbamate (**3**). The crude mixture of **3** was purified by column chromatography eluting with a gradient of ethyl acetate/hexane (20/80) to (30/70). Appropriate fractions were combined and concentrated to give a white solid which was then subjected to recrystallization. The product was dissolved in a minimum amount of methylene chloride and hexane was added dropwise until cloudiness was observed. Over 24 h, needle-like crystals formed providing compound **3** (21.3 mg, 34%). ^1^H NMR (500 MHz, CDCl_3_) δ 7.38 (s, 3H), 7.43–7.32 (m, 2H), 6.44–6.39 (d, 1H), 5.26 (d, *J* = 8.9 Hz, 1H), 5.14 (s, 2H), 4.91 (dt, *J* = 10.1, 7.9 Hz, 1H), 4.01 (dd, *J* = 8.8, 6.0 Hz, 1H), 2.98 (t, *J* = 8.8 Hz, 2H), 2.46 (dd, *J* = 13.3, 5.3 Hz, 1H), 2.18 (h, *J* = 6.8 Hz, 1H), 2.13–2.06 (m, 1H), 1.00 (d, *J* = 6.8 Hz, 3H), 0.95 (d, *J* = 6.8 Hz, 3H). ^13^C NMR (126 MHz, CDCl_3_) δ 204.44, 170.99, 152.17, 136.09, 128.61, 128.31, 128.13, 67.27, 64.17, 60.03, 42.28, 30.79, 29.71, 19.48, 19.18. HRMS (ESI): Calcd for (MH^+^) C_17_H_23_N_2_O_4_: 319.1652, found 319.1641.

Benzyl ((2S)-3-(4-hydroxyphenyl)-1-oxo-1-((2-oxocyclobutyl) amino) propan-2-yl) carbamate (**7**). The crude mixture of **7** was purified by column chromatography eluting with a gradient of ethyl acetate/hexane (20/80) to (30/70). Combined fractions were recrystallized from hot methylene chloride to afford compound **7** (14.6 mg, 24%) as small needle-like crystals: mp 154–155 °C. ^1^H NMR (500 MHz, acetonitrile-*d*_3_) δ 7.42–7.26 (m, 4H), 7.23–7.13 (m, 1H), 7.11–7.04 (m, 2H), 6.77–6.71 (m, 2H), 5.88 (dd, *J* = 12.9, 8.5 Hz, 1H), 5.08 (d, *J* = 12.8 Hz, 1H), 4.99 (d, *J* = 12.7 Hz, 1H), 4.80 (dddd, *J* = 20.0, 10.0, 5.9, 2.1 Hz, 1H), 4.25 (dtd, *J* = 17.5, 8.7, 5.3 Hz, 1H), 3.04 (ddd, *J* = 13.6, 7.8, 5.4 Hz, 1H), 2.96–2.81 (m, 1H), 2.79 (dddd, *J* = 14.1, 12.0, 8.3, 2.7 Hz, 1H), 2.15–1.91 (m, 1H). ^13^C NMR (126 MHz, CD_3_CN) δ 206.08, 171.17, 171.10, 155.73, 137.13, 130.43, 128.45, 128.27, 128.22, 127.86, 127.51, 115.09, 66.07, 63.71, 63.63, 56.31, 56.14, 41.39, 41.37, 18.33, 18.14.

Benzyl ((2S)-1-oxo-1-((2-oxocyclobutyl) amino)-3-phenylpropan-2-yl) carbamate (**9**). The crude mixture of **9** was purified by column chromatography eluting with a gradient of ethyl acetate/hexane (20/80) to (30/70). Appropriate fractions were combined and concentrated to give a white solid (94% HPLC purity) which was then recrystallized from hot methylene chloride to give a white needle-like crystal of compound **9** with 97% HPLC purity (30.2 mg, 49%): mp 100–101 °C. ^1^H NMR (500 MHz, CDCl_3_) δ 7.40–7.23 (m, 8H), 7.23–7.17 (m, 2H), 6.63 (NH, d, *J* = 7.6 Hz, 0.5H), 6.48 (NH, d, *J* = 7.9 Hz, 0.5H), 5.46 (t, *J* = 10.6 Hz, 1H), 5.07 (t, *J* = 3.3 Hz, 2H), 4.96–4.87 (m, 0.5H), 4.78 (qt, *J* = 7.9, 1.4 Hz, 0.5H), 4.45 (d, *J* = 7.0 Hz, 1H), 3.09 (d, *J* = 6.7 Hz, 2H), 2.98–2.83 (m, 2H), 2.38 (dd, *J* = 10.9, 5.2 Hz, 0.5H), 2.33 (dd, *J* = 10.3, 4.7 Hz, 0.5H), 2.03 (q, *J* = 9.6 Hz, 0.5H), 1.90–1.73 (m, 0.5H). ^13^C NMR (126 MHz, CDCl_3_) δ 204.79, 204.76, 170.84, 170.69, 156.03, 136.23, 136.23, 136.08, 129.40, 129.38, 128.76, 128.58, 128.26, 128.01, 127.14, 67.16, 64.03, 63.72, 55.97, 55.75, 42.16, 42.06, 38.63, 38.43, 19.35, 19.13. HRMS (ESI): Calcd for (MH^+^) C_21_H_23_N_2_O_4_: 367.1652, found 367.1646.

^1^H and ^13^C nuclear magnetic resonance (NMR) spectra for all final compounds **1**–**10** are included in the Appendix A.

### 2.2. In Silico Evaluation of Cyclobutanone Derivatives

A test library of approximately forty 2-aminocyclobutanone derivatives was prepared, including amide and sulfonamide derivatives as previously described [28]. The goal of this synthesized test library was to include broader molecular diversity while following druggability guidelines including Lipinski’s [33] and Veber’s [34] rules and including functionalities that could provide productive drug-target interactions. We, therefore, included a number of natural and unnatural amino acids to couple to the 2-aminocyclobutanone. Preliminary in silico screening of the series of synthesized and virtual 2-aminocyclobutanone derivatives revealed *N*-tosyl-D-phenylalanine cyclobutanone **6** as an early in silico hit against SARS-CoV-2 Nsp13 helicase, given the glide score of the α-(R)-stereochemistry cyclobutanone ketone hydrate (−5.72). This cyclobutanone hit is predicted to bind as the hydrate of the cyclobutanone carbonyl to the ATP binding site of the helicase, revealing the key binding interactions as shown in Figure 2.

We pursued analogs of tosyl-D-phenylalanine-cyclobutanone **6** as potential SARS-CoV-2 Nsp13 helicase inhibitors, selecting the set of ten total compounds that were synthesized (**1**–**10**), based on the results of docked **a**–**d** variants of compounds **1–10**, wherein **a** = (R)-stereochemistry and (unhydrated) cyclobutanone carbonyl; **b** = (R)-stereochemistry and ketone hydrate; **c** = (S)-stereochemistry and (unhydrated) cyclobutanone ketone; **d** = (S)-stereochemistry and ketone hydrate. Structural analogs missing from Table 1 (ex. **4a**–**4c**) did not dock and are therefore not included. Cyclobutanone-*N*-substituents explored included protected amino acid amides, including both (S)- and (R)-enantiomers of aryl, hydrophobic, and polar amino acids as well as unnatural amino acid derivatives of the original SARS-CoV-2 Nsp13 helicase hit. Various para-substituted arylsulfonamides (4-CH_3_, 4-CF_3_, 4-F, 4-Cl, 4-OCH_3_, and 4-H) *N*-benzyloxycarbonyl (Cbz) protected amino acid analogs, and cyclobutanone hydrate isosteres, including cyclopentanone, isopropanol, and cyclopropanol analogs, were explored. Our virtual high-throughput screening (HTS) methodology generated all the possible stereoisomers of each potential inhibitor, as well as ketone hydrates in addition to the cyclobutanone ketones. The lead optimization screening provided ten Nsp13 helicase hits with comparable or improved MM-GBSA estimated binding energies and Glide scores (Table 1). Glide score is a scoring function used to calculate the ligand binding free energy using OPLS3e force field (electrostatic and Van der Waals interactions were considered in the calculation).

All compounds that docked were predicted to bind to the ATP binding site of Nsp13 helicase through hydrogen bonding, pi-cation, and pi-pi stacking interactions with Gly287, Ser289, His290, Lys320, and Arg442. Interestingly, *N*-Cbz-L-Phe cyclobutanone **9** was identified as a helicase inhibitor with a Glide score of −5.039 for the α-(S)-aminocyclobutanone ketone (**9c**), whereas the epimeric *N*-Cbz-D-Phe α-(R)-aminocyclobutanone ketone (**2a**) was predicted to be more potent as based on the Glide score of −6.161. In silico hit compounds were prioritized for synthesis based on the binding energies, docking scores, and predicted synthetic challenges.

### 2.3. Biological Activities of Cyclobutanone Derivatives

We then demonstrated that cyclobutanone derivatives potently inhibit SARS-CoV-2 through viral inhibition assays to demonstrate their efficacy as potential antivirals, with promising in vitro antiviral activity and with little to no cytotoxicity at 5 µM and 50 µM (Table 1). The cell survival assay measured the ability of the tested inhibitors to prevent cell death caused by SARS-CoV-2 by calculating the number of living cells that escaped viral cell lysis. The assay data listed in Table 1 indicate the response of each inhibitor in cell survival assays after SARS-CoV-2 challenge. Higher host cell survival indicates better anti-SARS-CoV-2 activity of the inhibitors with minimal cytotoxic impact on the host cells.

Cell survival data is shown in Figure 3 against HUH-7 liver cells and fetal embryonic kidney cells, HEK 293, versus dimethylsulfoxide (DMSO) as a control. Out of the nine hits tested, tosyl-L-tryptophan cyclobutanone analog **4** showed the highest cell survival percentage indicating antiviral activity as well as potent anti-SARS-CoV-2 efficacy. Negative values in the chart would represent cell death due to the toxic effects posed by the inhibitors. When comparing the activities of tosyl-phenylalanine cyclobutanone **1** (L-isomer) and **6** (D-isomer) at both 5 µM and 50 µM, there was a significant increase in the activity of the L isomer over the D-isomer (Table 1). A similar trend was observed between Cbz-phenylalanine cyclobutanone epimers, where at 5 µM, the L-isomer **9** showed 41.5% cell survival versus only 24.2% survival in the presence of D-isomer **2.**

A plot of the dose response of in vitro antiviral efficacy for select cyclobutanone derivatives appears in Figure 4. Figure 5 shows the results of compound **4**, which has the best cytotoxicity profile from initial testing, against Omicron and Delta SARS-CoV-2 strains displaying inhibitory efficacy.

IC_50_ values of the top-performing compounds from initial testing showing significant inhibitory efficacies against wild-type (alpha strain, SARS-CoV-2, USA-WA1/2020) SARS-CoV-2 as shown in Table 2, along with IC_50_ values against Delta and Omicron strains for tosyl-L-trypophan cyclobutanone **4**. Tosyl-L-Trp-cyclobutanone **4** with the best cytotoxicity profile showed increased potency against now dominant Delta and Omicron variants of concern compared to the wild-type SARS-CoV-2. This suggests that, if these cyclobutanone derivatives are inhibiting Nsp13 helicase as suggested by in silico docking and modeling, then helicase may be a useful target in higher turnover variants.

The docked structure of cyclobutanone **4** in the ATP binding site of SARS-CoV-2 is shown in Figure 6. There are several productive H-bond/dipole attractions between, Gly287 and the cyclobutanone carbonyl oxygen, between Ser289 and amide carbonyl oxygen, and between Arg443 and sulfonamide oxygen. In addition, the ligand provides a hydrogen bond from the sulfonamide NH to the backbone carbonyl of Gly538. Lys320 side chain nitrogen lone pair forms bidentate interactions with the pi system of the indole of the ligand. The most interesting aspect is that the whole ligand participates in the binding and the binding pocket is highly complementary to the ligand pose.

## 3. Discussion

The COVID-19 pandemic has caused catastrophic effects on public health and the economy globally, and the loss of human lives in unprecedented numbers has been devastating. Fortunately, highly effective vaccines became available in late 2020, including against the B.1.617.2 (Delta) Variant [35], and vaccines continue to evolve to combat newer strains [36]. Oral medications, including molnupiravir, fluvoxamine, and Paxlovid, are effective in reducing mortality and hospitalization rates in COVID-19 patients [37]. Omicron Variants BQ.1.1 and XBB.1 have escaped neutralizing antibodies and are surpassing previous VOCs with antibodies generated post-monovalent or bivalent mRNA boosters [38], and these are now dominant strains in circulation [39]. Mathematical modeling predicts continued surges of new strains which are not hindered by acquired immunity, and which may have new distinct pathobiology and target systems rendering the previous immunomodulatory treatments ineffective [40]. Immunocompromised patients and patients suffering from long COVID or post-acute sequelae of COVID-19 (PASC) are dependent on antiviral treatments to keep the viral load below irreversible organ damage levels [41]. Thus, highly specific antiviral treatments targeting critical and highly conserved targets hold the key to continued progress against evolving lineages. The continued discovery and development of effective therapeutic agents are important for future COVID-19 variants and for future coronavirus events.

Our efforts in discovering potential drug candidate scaffolds against SARS-CoV-2 have focused on utilizing a multi-targeted, computer-aided drug design (CADD) approach. Drugs targeting essential viral proteins are preferred over antivirals targeting the host cellular mechanism to avoid or minimize adverse effects on human health, as they are designed to show higher specificity toward the virus. We identified a set of hits against SARS-CoV-2 Nsp13 helicase through a virtual HTS of synthesized cyclobutanone derivatives, initially screening against seven identified druggable SARS-CoV-2 target enzymes utilizing the Schrödinger suite and Molecular Operating Environment (MOE), then focusing on SARS-CoV-2 Nsp13 helicase exclusively. SARS-CoV-2 Nsp13 helicase is less studied, especially in comparison to the main protease (Mpro, or 3CLpro), and there are relatively few known inhibitors of SARS-CoV-2 Nsp13 helicase, and even fewer belonging to SAR, as summarized in Appendix A [42,43,44,45,46,47,48,49,50,51,52]. Inhibitors including repurposed drugs were shown to bind the SARS-CoV-2 Nsp13 helicase ATP binding site in recent studies [24,53]. Following Molecular Dynamics simulations (MDS) for in silico validation of target specificity, antiviral activity of *N*-functionalized 2-aminocyclobutanone hits and analogs was confirmed in vitro using a cell survival assay, and inherent cellular toxicity was assessed. In total, ten peptidomimetic cyclobutanone analogs were synthesized and showed promising antiviral activities with low toxicity on mammalian host cells in the cell survival assay. Among these, when evaluated for activity against SARS-CoV-2 WT strains, six compounds showed potent dose-dependent activity with IC_50_ ranging from 6–16 µM concentration (Figure 4). Furthermore, cyclobutanone **4**, which was least toxic against the Vero cell line and was moderately active against the alpha strain of SARS-CoV-2 (SARS-CoV-2, USA-WA1/2020) with an IC_50_ of 6.7 µM, showed efficacy against variants of concern Delta and Omicron, which are now dominant in circulation.

The six molecules tested against wild-type SARS-CoV-2 showed antiviral activity with different potencies (IC_50_ values). Overall, the molecules have low toxicity toward human cell lines in comparison to Amphotericin B, which is an FDA-approved compound for different disease indications. Although Nsp13 helicase is highly conserved among all strains, the difference in IC_50_ observed (10 to 20-fold; Figure 4 and Figure 5) for tosyl-L-Trp-cyclobutanone **4** was intriguing. This difference observed could be due to much higher replication rates (reproduction number: 9.5 Delta, 5.08 Omicron) in variants of concerns (VOCs), in comparison to the alpha strain (SARS-CoV-2, USA-WA1/2020) [54]. This may indicate that Nsp13 helicase is a more important target in higher turnover variants. The Omicron lineage is distinct from other lineages in that it has a reduced affinity towards lung parenchyma cells, which are targets for all other lineages [55]. Therefore, there is a need for follow-up testing with human cell lines to obtain a more clinically relevant assessment. Our data also suggest an urgent need to re-evaluate other reported inhibitors of Nsp13 helicase, such as cepharanthine, and test against current dominant VOCs, re-assessing their usability. It has been shown that other molecules, including EIDD-1931, PF-07321332, remdesivir, favipiravir, ribavirin, nafamostat, camostat, and aprotinin, target the replicase complex and show far more consistent activity against other variants [56]. Thus, our data may suggest that targeting Nsp13 helicase provides activity against the VOCs, as the helicase enzyme could be a bottleneck in the rapid replication of SARS-CoV-2 variants. We already have more potent molecules, specifically **2** (IC_50_ = 6.89 µM) and **10** (IC_50_ = 6.68 µM), tested against the wild type of variant. They have some inhibitory effect on Vero cell line growth, which could be due to a unique interaction with this epithelial cell line from an African green monkey. They, therefore, need to be further tested in an alternate system with human cell lines including transgenic A459 cells expressing ACE2 as well as TMPRSS2, as they have less nonspecific effect with human cell lines.

Although docking and modeling strongly suggest that SARS-CoV-2 Nsp13 helicase is the target of these cyclobutanones, and binding is predicted to occur at the ATP binding site, further studies will be required to validate the ligand-target specificity using in vitro enzymatic assays. Targeting an ATP binding site includes the concern about selectivity, as for the multitude of kinase inhibitors, mostly targeting the ATP binding site, or even non-ATP off-targets [57]. Furthermore, ATP competitive inhibitors must be able to compete with the ATP concentrations in the cell, which are around 1–10 mM in human cells [58]. Nevertheless, the great success of kinase inhibitors targeting the ATP binding site is encouraging [59]. Further optimization of potent and selective inhibitors with improved efficacy against SARS-CoV-2 enabled by CADD against SARS-CoV-2 Nsp13 helicase is in progress. Herein, we report a new druggable cyclobutanone scaffold exploiting an essential but relatively less explored target in the SARS-CoV-2 proteome that is nevertheless critical for viral replication. As Nsp13 helicase is the most conserved protein among all *Coronaviridae,* which are known to be responsible for two major previous pandemics (SARS & MERS), these early inhibitors could serve as drug leads toward future lines of defense in treating outbreaks from this notorious family.

## 4. Methods and Materials

### 4.1. Organic Synthesis Methods

All reagents, from Sigma Aldrich (St. Louis, MO, USA) were used without further purification, unless otherwise noted, and solvents were distilled before use. For column chromatography, RediSep^®^ silver silica gel flash columns (Teledyne Isco, Lincoln, NE, USA) were utilized, and aluminum-backed silica gel 200 µm plates (Sorbent Technologies, Inc. Norcross, GA, USA) were used for thin-layer chromatography (TLC). 1 H (proton) NMR spectra were obtained at 500 MHz and 13C were obtained at 125 MHz using a 500 MHz Bruker spectrometer (Bruker Corporation, Billerica, MA, USA), with tetramethylsilane (TMS) as the internal standard. NMR spectra were processed using the Mnova NMR software program provided by Mestrelab Research (Santiago de Compostela, Spain). The purity of all compounds that were assayed was confirmed to be ≥95% as determined by Agilent 1100 high-performance liquid chromatography (Agilent Technologies, Inc. Santa Clara, CA, USA) utilizing a mobile phase A, comprised of 5% acetonitrile in water, and a mobile phase B, comprised of 0.1% trifluoroacetic acid (TFA) in acetonitrile, employing a gradient of 60% B increasing to 95% over 10 min, holding at 95% B for 5 min, and then returning to 60% B, and finally holding for 5 min. High-resolution mass spectral (HRMS) data were obtained at the Integrated Molecular Structure Education and Research Center (IMSERC, Northwestern University, Evanston, IL, USA) on an Agilent 6210A TOF mass spectrometer (Agilent Technologies, Inc. Santa Clara, CA, USA) in the positive ion mode coupled to an Agilent 1200 (Agilent Technologies, Inc. Santa Clara, CA, USA) series high-performance liquid chromatography (HPLC) system. Data was processed using MassHunter software version B.04.00 (Agilent Technologies, Inc. Santa Clara, CA, USA).

### 4.2. In Silico Methods

Protein/Receptor structure models. The crystal structures of NSP13 (Nsp13 Helicase; PDB ID: 6XEZ), Nsp3 (PDB ID: 6W02), Nsp9 (PDB ID: 6W4B), and Nsp15 (PDB ID: 6VWW) of SARS-CoV-2 were obtained from Protein Data Bank [60]. The crystal structures of SARS-CoV-2 main protease (Nsp5) in complex with Z44592329 (ID: 5r83) were obtained from the Protein Structure Database of Europe [61].

Protein (Receptor) preparation. Protein preparation was performed using the Protein Preparation Wizard in Maestro [62]. Crystallization ions present in the crystal structure, including calcium and chloride, were deleted. Crystallization ligands present in the active site were not deleted as they were utilized for grid generation. All water molecules with less than three hydrogen bonds (Sample water orientation) with the receptor or the ligand were deleted. Structure preparation of all seven receptor models developed was performed using Wizard in Maestro. Each of the corrected receptor complexes was optimized with minimized hydrogens and then minimized using the OPLS3e force field [62]. Then, MD simulations (MDS) of the receptor models were performed under physiological conditions (0.15 M NaCl) by solvating the models in water. A simulation (water) box covering the entire receptor model was created with a 10 Å buffer space. The MD simulation was run for 20 ns at 300 K at standard pressure (1.01325 bar). Interacting residues were studied via a target complex trajectory analysis performed using the Desmond software [62].

Ligand library preparation. Ligand preparation was performed on LigPrep, where potential structural variations of ligands were generated, reactive species were eliminated and ligands were optimized. Optimization was performed utilizing the OPLS3e force field. Protonation states of ligands at pH 7.0 ± 2.0 were taken into consideration and an EPIK minimization was performed on all possible protonated species. Tautomers for each ligand were created while retaining the chirality combinations, with a maximum of 32 structures for each ligand.

Receptor grid preparation. Active sites of the receptors were located, and the size was estimated using COACH analysis. A grid with appropriate shapes and properties of each receptor was created for more accurate and refined screening and to avoid the exclusion of possible active compounds. Binding pocket residues were predicted using the centroid of the COACH, through which grids were generated using default values of protein atom scaling (1.0 Å) within a cubic box and ligand docking set to a length of 20 Å. OPLS3e force field was employed in grid generation. During the grid generation, the Receptor Grid Generation wizard of the Glide module intrinsically determined rotatable groups. COACH predicted binding site residues of each target were represented as Nsp13 Helicase; extracted from a replicase complex model (PDB ID: 6XEZ), the nucleic acid binding site comprises amino acid residues 177, 178, 179, 180, 181, 197, 214, 309, 310, 311, 334, 335, 336, 337, 338, 339, 408, 409, 410, 411, 412, 413, 414, 485, 486, 516, 534, 554, and 560.

High-Throughput Virtual Screening (HTVS). The HTV screening was performed in the Glide (a Grid-based ligand docking that utilizes energy calculations) module of the Schrödinger suite [62]. Default parameters, including ionization states at defined pH, and Epik state penalties (for ionization and/or tautomeric states at physiological pH) were taken into consideration. The scaling factor was fixed at a default of 0.8 and a 0.15 partial charge cut-off was selected. The force field used for docking was OPLS3e. An HTVS ligand docking was performed first, followed by SP and XP docking on the top 10% scoring hits from the appropriate previous step. The XP docking is an accurate analysis that eliminates false positives through a more rigorous scoring function than the HTVS. The binding affinity of the ligand is determined by the Glide score. The greater the XP GScore, the better the estimated affinity of the hit toward the protein target. Binding free energies of the best-docked ligand-receptor complexes were calculated using molecular mechanics (MM) force fields and implicit solvation was performed using the in-built molecular mechanics/generalized Born surface area (MM-GBSA) method of the Schrödinger virtual screening platform [62]. The binding energy calculation used the following equation:ΔG = E_complex(minimized) − (E_ligand(minimized) + E_receptor (minimized))

The test ligands were ranked based on the estimated binding free energy for the corresponding ligand-receptor complexes.

Molecular Dynamics Simulation (MDS). The stability of the ligand-receptor complex and the energetic strain on the docked ligand during the induced fit (flexible binding site) docking screening were measured. The increased strain energy of the ligand could result in immediate fly-off (<10 ns). Our integrated HTVS and MM-GBSA screening pipeline [10,11,26,63] was validated via a 20 ns MD simulation of top hits from each target pool following the method stated in the protein/receptor preparation section. When the ligand/hit experienced a major conformation change, it flew off immediately from the receptor binding site. In that case, a 20 ns simulation of the free receptor was performed until it reached a stable state, and the ligand screening was repeated with the resulting minimized receptor.

### 4.3. In Vitro Methods

Anti-SARS-CoV-2 testing. Anti-SARS-CoV-2 testing was performed in a BSL-3 Laboratory (Howard T. Rickett Regional Biocontainment Laboratory): Cells and Virus Used. Human alveolar basal epithelial A549-ACE2 cells and SARS-CoV-2 [novel coronavirus (nCoV)/Washington/1/2020] provided by N. Thornburg (CDC) via the World Reference Center for Emerging Viruses and Arboviruses (Galveston, TX, USA) and from BEI Resources. Variants of concern were obtained from BEI resources. Delta Variant (BEI Cat.ID. NR-55671) Isolate hCoV-19/USA/MD-HP05285/2021 (Lineage B.1.617.2) was contributed by Andrew S. Pekosz. Omicron Variant (BEI Cat.ID. NR-56461) Isolate hCoV-19/USA/MD-HP20874/2021 (Lineage B.1.1.529) was also contributed by Andrew S. Pekosz.

In the evaluation of viral inhibition of test compounds through Spike Immunohistochemistry (IHC) assay, all SARS-CoV-2 infections were performed in biosafety level 3 conditions on the Cells in DMEM +2% FBS. For the preliminary selection of hits, cells were pre-treated with test compounds for 2 h with 2-fold dilutions beginning at 50 µM in triplicates for each assay. To enumerate the IC50 or percent inhibition, an identical treatment was performed with 10-fold dilutions beginning at 50 µM. A549-ACE2 cells were infected with an MOI (multiplicity of infection) of 0.5 in media containing the appropriate concentration of drugs. After 48 h, the cells were fixed with 10% formalin, blocked, and probed with mouse anti-Spike antibody (GTX632604, GeneTex) diluted 1:1000 for 4 h, rinsed, and probed with anti-mouse-HRP for 1 h, washed, then developed with DAB substrate 10 min. Spike-positive cells (n > 40) were quantified by light microscopy as blinded samples.

Mammalian cytotoxicity testing. HEK293 and HUH7 cells were maintained in Dulbecco’s modified Eagle’s medium supplemented with 10% Fetal bovine serum and 1% Penicillin–Streptomycin (Gibco) and maintained in filter cap cell culture flasks at 37 °C in a humidified atmosphere of 5% CO_2_. HEK293 and HUH7 cells were plated in separate plates at a seeding density of trypsinized 5000 cells/100 µL/well in a black clear bottom cell culture grade 96 well plate. The plates were incubated for 24hrs at 37 °C and humidified at 5% CO_2_ after adding different test compounds, along with Amphotericin B as a positive control, and DMSO vehicle (same as sample volume 1 µL) as the negative control. Compounds were dissolved in cell culture grade DMSO (stock concentration of 10 mM; 100× diluted to 100 µM highest concentration). The highest concentrations (100 µM) were 1:2 serially diluted (10 dilutions) and each series was performed in triplicates. The incubated plates were stained with Hoechst 33342 (Thermo Fisher Scientific, Carlsbad, CA, USA) dye (6 µM final concentration), incubated for 24 h at 37 °C, and humidified with 5% CO_2_ for 30mins. The cells were then imaged on the ImageXpress Pico High-Content Imaging System Microscope from molecular devices using a 4× plan fluor objective (4× Plan Apo Lambda Nikon air objective lens with a camera binning of 2 and a pixel size of 3.367 µm × 3.367 µm) with bright field and LED illumination capture DAPI channels by Sony CMOS inbuilt camera. Furthermore, the imaging data were processed with automated data programs with pre-configured analysis protocols and nuclei counts were documented. Captured images were analyzed to measure the integrated nuclear fluorescence area of each well. The CRC software algorithm computes the cell perimeter shape and the location of the center of mass of each nucleus from fluorescent images and three-dimensional images of the nuclei were assembled from multiple z-plane scans. this produces calculated average intensity as well as the total area in a well. Integrated nuclear fluorescence area for each well was used to estimate cell death and calculate CC_50_ of test compounds using Prism software version 9 (GraphPad) with normalized values on the *y*-axis and log transformed drug concentrations on the *x*-axis, performing Nonlinear regression (curve fit) and selecting inhibition curve (IC_50_) calculation module

Testing against SARS-CoV-2 Variants. The testing method was the same as mentioned for the Wild type (Alpha strain, SARS-CoV-2, USA-WA1/2020). The drug dilutions used were 10-fold lower, i.e., starting from 10 µM and diluting 50% serially.

Values of half-maximal inhibitory concentrations (IC_50_), effective concentration (EC_50_), and cytotoxic concentration (CC_50_) were calculated by nonlinear regression using GraphPad Prism 8.0 (GraphPad Software, Inc., La Jolla, CA, USA). All data are expressed as the mean ± SD of triplicate assays.

## Data Availability

The data presented in this study are available in this article and in the Appendix A.

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
