# Peer review of "In Silico Binding of 2-Aminocyclobutanones to SARS-CoV-2 Nsp13 Helicase and Demonstration of Antiviral Activity"

_ijms, 2023, doi:10.3390/ijms24065120_

Round 1
Reviewer 1 Report
Overall, the work has merit, but there appears to be a lot of confusion on the relevance of the assays performed, which are indicating antiviral activity, and the unevidenced assumption that the compounds do bind the viral helicase. The authors say so themselves at line 283: “Further studies are required to validate the ligand-target specificity using in-vitro enzymatic assays.”
As it stands, the manuscript contains too many misleading statements and assumptions regarding the activity of these molecules, and the rationale behind the study, including the molecular modelling studies, has not been explained to an acceptable standard. In addition, the compounds made are peptide-mimics, and this only appears very late in the draft: a better description of this class of molecules must be made in the introduction, and the reason to claim they inhibit the helicase (why shouldn’t they inhibit a viral protease instead?) must be clearly and logically explained. Furthermore, most of the assays presented lack any statistical analysis: this is another major flaw of the work, and it must be addressed for publication to be considered.
Introduction, lines 76-77: the description of the helicase protein domains should be supported by at least a figure, and the different potential binding sites at the level of the protein should be shown and detailed.
In the introduction, it is highly unclear why the authors thought their cyclobutanone compounds would be active against the helicase enzyme: this should be clearly explained in the introduction. A it stands, the introduction sounds like a series of separate points which are not connected to one another: the activity of the compounds on different proteases, then the SARS-CoV-2 helicase, then the preparation of a library of cyclobutanones for unexplained reasons, then the computational studies of these compounds on the helicase. The different parts need to be logically connected to one another, and the rationale of the study needs to be clearly explained.
Reported inhibitors of the SARS-CoV-2 helicase should be summarised, and their chemical structures should be shown in a figure.
Figure 1: an additional overview figure should be added here, in order to show where on the helicase in which site/domain) the compound is predicted to bind.
Why was compound 6 docked in the first place? Was a selection of in-house molecules docked in the helicase (in which site? The ATP binding site of the helicase in only mentioned at line 127: this needs to be clarified much earlier in the text) to find potential good binders? If so, this should be explained, and also a brief description of the in-house library evaluated with docking simulations should be given.
Advantages and disadvantages of targeting the ATP binding site of the protein (e.g., potential toxicity) should be discussed.
Line 175, the statement “can serve as potent competitive SARS-CoV-2 helicase inhibitors through viral inhibition assays” is wrong: competitive inhibition of an enzyme cannot be demonstrated with the cell-based antiviral assay performed. This assay only indicates inhibition of the viral replication, but it does not give any indication of the actual target of the compounds: the target may or may not be the helicase. This misinterpretation of the data must be addressed.
Line 181, another incorrect statement that must be amended “The assay data listed in Table 1 indicate the dose response of each inhibitor in cell viability and relative cytotoxicity.” The data shown in Table 1 correspond to only 2 test concentrations: these are not dose-response curves! Figure 3 presents dose-response curves, but not Table 1.
Any statistical analysis of the antiviral and cytotoxicity data shown in Table 1, Figure 2 and Figure 3 is missing: this appears to indicate that the compounds were only tested once, not in triplicate. Repetitions of the assays performed must be added, and the statistical analysis of the data must also be added.
Figure 4: the error bars are huge, so the figure does not support the statement made by the authors that the compound is “displaying statistically significant inhibitory efficacy”: more data need to be collected to reduce the error bars, before this statement can be made.
Line 213: The “CADD approach” claimed by the authors is unclear. There is no evidence that these compounds actually inhibit the viral helicase, as only cell-based antiviral assays have been performed, and the CADD approach adopted has not been clearly and logically detailed. Please see my comments above.
Line 222 and 233: another assumption that cannot be made. The authors claim that the activity of the compounds “demonstrates helicase is a more important target in higher turnover variants.”. The authors have demonstrated that the compounds appear to inhibit the viral replication, but they have not reported any data that confirm inhibition of the helicase. They only have predicted the binding to the helicase with molecular modelling studies. This statement therefore is misleading and, in the absence of experimental data on the target, it cannot be made.
Table 2: standard deviation values must be added. Most of the data presented in this study do not show any statistical analysis: why is this? This must be addressed.
Line 280: green highlight.
Several references must be added to the experimental methods part, especially for the in vitro antiviral assays.
Reviewer 2 Report
This manuscript reports the initial characterization of a group of aminocyclobutanone compounds that serve as inhibitors of helicase (nsp13) of SARS-CoV-2. This is reported as the first of its kind. Furthermore, these compounds are more effective against Omicron than delta variants of the virus. The authors touted its potential to serve as an antiviral against future SARS-CoV oiutbreaks.
This is a rather preliminary report of the anti-SARS-CoV drug development. The salient finding is that those compounds that are developed later tend to be more efficacious against Omicron than delta variants; this finding was interpreted to mean that further developments of this type of compounds have potential to be effective against future variants. This is a very brave but un-substantiated speculation. Provide some evidence or tone down the arguments.
Overall, this is just the beginning of a series of steps in silico drug development for SARSCoV-2 More efforts are needed to make it worthwhile for publication.
Reviewer 3 Report
Comments on ijms-2206220
In this study, the author has studied "Antiviral Activity of 2-Aminocyclobutanones Targeting SARS-CoV-2 Helicase." The present manuscript is the withdrawn manuscript from "Loyola University Chicago" in 2020. In the era of vaccines, the present manuscript lost its novelty. The authors did not report strong evidence regarding the importance of the topic. The presentation of the methodology is somewhat confusing, and the readability of the discussion also needs major improvement. The English language used in the manuscript needs some improvements as some punctuation and grammatical mistakes are present. Experimental designs required more clarity. Moreover, research models are not discussed in an understandable manner, reflecting that the author needs a more comprehensive way of thinking.
Specific comments:
1. The Abstract needs to be critically revised. Please add a clear objective of the study and add more results with concluding remarks.
2. Please add more strong keywords and avoid the words already used in the title.
3. Page 2, line 36-37: "This fatal and highly contagious coronavirus has caused over 660 million COVID-19 cases globally, resulting in over 6.7 million deaths." Please add the access date because numbers could be changed day by day.
4. Page 2, line 43: "Targeting the same proteins increases the chance of cross-resistance i.e. resistance-conferring…" Please avoid the words' i.e., etc.' in scientific manuscripts.
5. Page 3, line 66-82: "Helicase, designated as nonstructural protein 13 (Nsp13) of SARS-CoV-2… was demonstrated to have modest (IC50 = 0.4 mM) anti-SARS-CoV-2 activity." Please rearrange this paragraph and shift to above the upper paragraph (as the second paragraph).
6. Page 3: The whole introduction section looks general and has a few limitations. The authors are advised to revise the introduction section carefully and add some data on the therapeutics of COVID-19. Please also report that what is the importance of 2-aminocyclobutanones in the era of vaccines. Many prestigious companies have developed successful vaccines, so in silico studies have very little importance. The introduction does not have a proper problem statement and research gap. The authors have to give some strong evidence related to the present topic.
7. Page 3: What is the research gap and novelty of the present study?
8. The authors are advised to add the structural homology of 'nsp helicase.'
9. In the "Results and Discussion" section, where is the discussion? This section needs revision. This needs professional English editing, and please revise it carefully to make it standard. Please focus on the main topic during the discussion. An excellent discussion contains an accurate statement of the results, the relevance, and importance of the results, suitable comparisons to similar studies, alternative explanations of the findings, known limitations, and suggestions for future research.
10. The authors mentioned SARS-CoV-2 helicase; it is better to mention SARS-CoV-2 nsp13 helicase and follow this trend throughout the manuscript.
11. The authors are advised to follow the Author's instruction of IJMS.
12. Authors are advised to proofread the manuscript to overcome grammatical mistakes.
13. Authors are advised to revise headings and subheadings.
14. Most of the references are outdated; please revise them and add updated data.
Reviewer 4 Report
The manuscript: Antiviral Activity of 2-Aminocyclobutanones Targeting SARS-CoV-2 Helicase by Prakasha Kempaiahb, Daniel P. Becker and collaborators is a very important research that can open new development of covid 19 treatments and should attract the interest of researchers in multiple scientific areas. I think that chemical modification have a good chance to result in more active products in the future.
The major point of this research is to develop a drug against covid based on a region of the virus that is less prone to mutation and can in addition help at the same time with the development of a broader vaccine to new mutants. This addresses an important gap in the field. The cyclobutane key part of the molecule can be easily modified and can serve as a template for other type of viruses. The future in vivo experiments in different animals will be a critical step in the development of a new drug. The references are appropriate at the current time.

Round 2
Reviewer 2 Report
The primary merit of this manuscript lies in the potential in silico synthesis of the new type of inhibitors for the helicase of SARS-2 virus. The successful synthesis of these inhibitors appears to be convincing after the authors have included appropriate controls in the assays in the revised manuscript. These controls removed my reservations about the validity of the assays. Although there are still some over-interpretation (e.g., the claim that helicases are the most powerful components of the SARS-2 replication enzymes), the overall findings of this manuscript are good and justifiable.
Reviewer 3 Report
The authors have carefully addressed all the comments. So, the manuscript should be accepted in its present form.